# Study on Mechanical Properties of Basalt Fiber Shotcrete in High Geothermal Environment

**DOI:** 10.3390/ma14247816

**Published:** 2021-12-17

**Authors:** Yueping Tong, Yan Wang, Shaohui Zhang, Yahao Chen, Zhaoguang Li, Ditao Niu

**Affiliations:** 1College of Materials Science & Engineering, Xi’an University of Architecture and Technology, Xi’an 710055, China; tongyueping@163.com (Y.T.); chenyahao619@163.com (Y.C.); 2State Key Laboratory of Green Building in Western China, Xi’an University of Architecture and Technology, Xi’an 710055, China; niuditao@163.com; 3College of Civil Engineering, Xi’an University of Architecture and Technology, Xi’an 710055, China; zhangshaohui999@126.com (S.Z.); lizhaoguang@xauat.edu.cn (Z.L.)

**Keywords:** high geothermal environment, shotcrete, basalt fiber, compressive strength, flexural strength, stress-strain constitutive

## Abstract

With the development of infrastructure, there are growing numbers of high geothermal environments, which, therefore, form a serious threat to tunnel structures. However, research on the changes in mechanical properties of shotcrete under high temperatures and humid environments are insufficient. In this paper, the combination of various temperatures (20 °C/40 °C/60 °C) and 55% relative humidity is used to simulate the effect of environment on the strength and stress–strain curve of basalt fiber reinforced shotcrete. Moreover, a constitutive model of shotcrete considering the effect of fiber content and temperature is established. The results show that the early mechanical properties of BFRS are improved with the increase in curing temperature, while the compressive strength at a later age decreases slightly. The 1-day and 7-day compressive strength of shotcrete at 40 °C and 60 °C increased by 10.5%, 41.1% and 24.1%, 66.8%, respectively. The addition of basalt fiber can reduce the loss of later strength, especially for flexural strength, with a increase rate of 11.9% to 39.5%. In addition, the brittleness of shotcrete increases during high temperature curing, so more transverse cracks are observed in the failure mode, and the peak stress and peak strain decrease. The addition of basalt fiber can improve the ductility and plasticity of shotcrete and increase the peak strain of shotcrete. The constitutive model is in good agreement with the experimental results.

## 1. Introduction

A growing number of railway and highway tunnels are being constructed imminently in the Sichuan–Tibet region of China with the advancement of the “The belt and The Road” strategy. The engineering of high geothermal phenomena occurs more frequently due to geological structure, magmatic activity, groundwater activity, and other factors. This special environment leads to adverse effects on building procedure, while the tunnel construction deteriorates in long-term service [1]. Therefore, the high geothermal problem for tunnels is an important engineering problem to be solved.

Generally, one of the main functions of high temperature is considered to be accelerated cement hydration, resulting in the rapid but highly non-uniform formation of a structure. This is beneficial to the development of early strength, although the increase in defects in concrete also causes a decrease in homogeneity, and the coarsening of the pore structure degrades mechanical properties [2,3]. The early researches of the author’s group showed that the accelerated hydration due to accelerated or high temperature curing can promote the formation of more hydration products, especially ettringite (AFt), and improve the early strength of concrete, while a strength reduction at later age was attributed to the decomposition of Aft [4]. Meanwhile, previous researches show that two different high geothermal environments, made up of high temperatures combined with a high or low humidity, also have various effects on shotcrete properties. The results of Tang et al. [5] suggested that under a high humidity of 90%, the strength reduction of concrete primarily occurrs when the curing temperature is above 60 °C. Liu et al. [6] studied the strength of shotcrete at a low humidity of 35%, and found that the temperature threshold of strength changes is 40 °C. Yin [7] systematically studied the effect of humidity on mechanical properties of concrete, and found that, compared with an environment with relative humidity (RH) of 95%, the strength at RH of 25% continually decreases with the increase in surrounding rock temperature. Generally, excessive curing temperature is destined to make shotcrete deteriorate, and a low humidity can accelerate the process.

Hence, many scholars have concentrated on improving the high-temperature properties of shotcrete and found that the incorporation of fiber is one of the effective methods [8,9,10,11]. Cui et al. [12] using steel and basalt fibers to alleviate the performance of concrete in an environment of 100 °C with RH of 95%, and found that fiber can improve compressive strength and splitting tensile strength, especially steel fiber. Zhang et al. [13] studied the effect of polypropylene fiber on shotcrete in a high-temperature diversion tunnel, and found that polypropylene fiber and polyacrylonitrile fiber can improve early tensile strength. The results of Wang et al. [14] showed that the cooperation of steel or polypropylene fiber can reduce the loss of compressive strength to 3.7%, while incorporation of them alone made strength decrease by 33.8% and 33.4%, respectively. Zhou et al. [15] discovered that, at high temperature and low humidity, the strength loss of shotcrete can be reduced by adding 0.1% basalt fiber and 5% silica fume. Moreover, some plant fibers also were anticipated to solve the defect, but this still needs further research [16,17].

In sum, previous findings have confirmed the importance of temperature and humidity in tunneling. However, the application of lining concrete in the compound environment of high temperature and relatively low humidity is still insufficient; the low humidity environment will aggravate the shrinkage and cracking of concrete and damage the lining structure, which will have a great influence on the strength of concrete and the coordinated deformation ability of lining and surrounding rock, and the effect of common methods used to optimize properties of concrete is also unclear under this environment. Meanwhile, the only strength variation is dissatisfied with the deformation design of the lining structure, and it is necessary to further explore the relationship between concrete deformation and constitutive. Therefore, in this paper, a similar environment was established to simulate the high geothermal environment while basalt fiber was used to alleviate the degradation of the tunnel structure. Compressive strength, flexural strength, and the axial compressive stress–strain curve of basalt fiber reinforced shotcrete (BFRS) in a high geothermal environment were studied and a constitutive model was proposed.

## 2. Materials and Methods

### 2.1. Raw Materials

Ordinary Portland cement (OPC) was used as binder. Its chemical cpmposition and physico-chemical properties are shown Table 1 and Table 2, respectively. The river sand and the break stone were adopted as fine and coarse aggregates, respectively. Its properities is shown in Table 3. The superplasticizer was liquid polycarboxylate superplasticizer, in which water-reduction rate and dosage were 25% and 1.0 wt.%, respectively. An alkali-free liquid accelerator with a dosage of 4.0 wt.% was adopted, with initial and final setting times of 200 s and 447 s, respectively, and their chemical compositions are shown in Table 4. Basalt fiber (BF) was used to reinforce shotcrete, its properties are presented in Table 5.

### 2.2. Mix Proportions Design and Test Methods

Five types of shotcrete were prepared in this experiment, their mixture proportions are presented in Table 6. The BF contents in different concretes were 0, 0.1 wt.%, 0.2 wt.%, 0.3 wt.%, and 0.4 wt.% of binder, respectively. The specimens were prepared according to the Standard of China (CECS 13: 2009) [16].

According to the measured temperature and humidity of the Sichuan–Tibet Line Sang Chu Ling Tunnel [17], the temperature range and RH of the actual tunnel are approximately 40 to 60 °C, and 55 ± 5%. So, two temperatures of 40 ± 2 °C or 60 ± 2 °C, combined with a constant RH of 55 ± 5% were set to simulate a high geothermal environment. The specimens were demolded after curing at ambient temperature for 1 day, then were cured at two high geothermal environments and a standard environment with a temperature of 20 ± 2 °C and RH of 95 ± 5%, respectively. After being cured to special age, the specimens were moved out and used in the tests of compressive strength, stress–strain curve, and flexural strength. According to the Standard of GB/T 50081-2019, three specimens were poured in groups of 100 mm × 100 mm × 100 mm, 100 mm × 100 mm × 300 mm, and 100 mm × 100 mm × 400 mm, respectively [18]. In addition, the different specimens were marked as X-BFY, in which the X represents curing temperature and Y is the content of BF.

An electro-hydraulic press with a capacity of 1000 kN was used for the compressive strength test. The stress–strain test was carried out on a universal material testing machine. Two displacement sensors were arranged on the adjacent surfaces of concrete. A constant displacement rate of 0.1 mm/min was set to control the loading. The test site is shown in Figure 1.

Flexural strength was accessed by four-point bending test and the result was calculated according to Formula (1).
(1)$$f_{c}=\frac{P_{0.1}\cdot L}{b\cdot d^{2}}$$
where *f*_c_ is the flexural strength of shotcrete, MPa; *P*_0_._1_ is the peak load of the specimen, KN; *L* is the span between supports of beam specimens; *b* is beam width; *d* is beam height.

## 3. Results and Discussion

### 3.1. Compressive and Flexural Strength

The influence of curing temperature on the compressive strength of shotcrete is shown in Figure 2a. With the curing temperature rising, the change in flexural strength of shotcrete without or with 0.3% BF is presented in Figure 2b. It can be seen that high temperature is beneficial to the development of the flexural strength. In the condition of maintenance at 40 and 60 °C, the 7 d to 60 d flexural strength of shotcrete without fibers increased by 8.6% to 29.3% and 2% to 18.6%, respectively. When 0.3% BF is added, the flexural strength of BFRS increased a little in contrast to shotcrete without incorporation of fibers, particularly in the condition of maintenance at high temperature. In the case of maintenance at 40 °C and 60 °C, the 7 d to 60 d flexural strength of BFRS increased, respectively, by 11.9% to 37.3% and 19.9% to 39.5%.

High temperature could accelerate the cement hydration and the formation of the gel network, leading to an increase in early age mechanical properties of the shotcrete. However, excessive curing temperature is often accompanied by over-quick hydration. This implies that the stable structure of the matrix is formed before the hydration products spread evenly, and therefore many defects in micro scale exist in the matrix. Meanwhile, due to quick evaporation of water, a larger shrinkage stress arises. Under the combined action of the two, the strength of shotcrete may deteriorate. When 0.3% BF is incorporated, the effect of temperature on the flexural strength of shotcrete produces a slight deviation, which will be discussed below.

Adding 0.3% of BF will decrease the quality and increase the inner shortcomings of shotcrete, and negatively affects the process of the compressive strength. Incorporating fiber into shotcrete is useful to keep the shotcrete from splitting and decreases the degradation of flexural strength that results from the high temperature environment [19,20].

Then, the curing temperature of 60 °C, at which shotcrete performance deteriorates the most, is selected to study the effect of fiber content and the results are shown in Figure 3. Figure 3a shows the compressive strength of BFRS in the curing condition of a high geothermal environment at 60 °C. It was found that the incorporation of BF contributes to a negative effect on the development of early compressive strength, whereas it can improve the deterioration of concrete compressive strength during long-term maintenance at high temperature. Compared to the concrete without fiber, the 1 d and 7 d compressive strength of concrete decreases by 7% to 29.9% and 9% to 17.5%, respectively, when the incorporated amount of fiber is 0.1% to 0.4%; during maintenance at about 28 d to 60 d, the compressive strength of BFRS, incorporating an amount of 0.1% to 0.3%, increases 33.3% to 41.1% and 7% to 14.1%, respectively, but superabundant fiber has an adverse effect on development of compressive strength. Excessive fiber has a greater risk of agglomeration, resulting in strength reduction of concrete.

Figure 3b shows the flexural strength of BFRS in the curing condition of a high geothermal environment at 60 °C. It can be found that, when the fiber content is 0.1 to 0.3%, the concrete flexural strength increases with the increase in fiber content, but when it exceeds 0.3%, the strength decreases. At 7 d age, the flexural strength of BFRS increases 12.4% to 51.9% compared with concrete without BF. However, the flexural strength of BFRS decreases so that its range is 2.6% when the incorporated amount of BF is 0.4%. With the increase in curing age, the 60 d flexural strength of BFRS increases 3.1% to 28.3% when the incorporated amount of fiber is 0.1% to 0.3%. However, under the combined action of a high temperature and accelerating agent, the hydration of cement is over-quick and many defects are produced in the matrix of shotcrete [21]. The addition of moderate BF could restrain the development of cracks and decrease the reduction of concrete performance due to high temperatures. However, excessive fiber is always followed by a great risk of agglomeration, which may be due to the fact that a greater amount of BF is not easy to disperse and can be clustered together, resulting in limited performance improvement of concrete [22,23].

### 3.2. Uniaxial Compressive Stress-Strain

#### 3.2.1. Compression Failure State

The destruction form of shotcrete under axial compression is presented in Figure 4. As shown in Figure 4a,c,e, several large cracks, even small pieces, appear on the surface of the specimen without BF; when cured at 40 °C, the integrity of concrete is seriously damaged. When the concrete was cured in a standard environment, several fine cracks appear and the edges and corners fell off, but the integrity was good. After the shotcrete with high temperature curing reached the maximum load, it made a cracking sound and dropped rapidly in the process of falling. Under the effect of BF, although a spalling phenomenon was observed on the side, the damage degree was slighter than the former.

Figure 4a–d indicates that the effect of BF content on damage degree of shotcrete cured at 60 °C. The less BF content is added, the more serious the damage of concrete is, and a slight burst sound appears in the loading process. Small cracks gradually change to large cracks, until the middle part of the split, resulting in incomplete morphology. With the increase in BF content, the destruction form gradually develops from a large crack to small cracks and there is no large area of crushing and falling off, the degree of damage is slight. Meanwhile, it is found that, as more BF is incorporated, the loading time of BFRS is longer and the time taken for large cracks to appear is prolonged.

#### 3.2.2. Effect of Curing Temperature on Stress-Strain Curve of Shotcrete

The stress-strain curve of shotcrete at different curing temperatures is shown in Figure 5. At the age of 7 days, the rising slope of the stress-strain curve of shotcrete under high temperature curing increases, the peak point shifts to the left and the decreasing rate is faster than that of standard curing. From 28 to 60 days, the peak point of shotcrete cured at high temperature did not move up obviously, but the descending section showed a buffer stage and the curve was prolonged. With the addition of BF, the peak point of BFRS at high temperature curing is lower than that of standard curing BFRS, However, the stress-strain curve of concrete under long-term curing is prolonged, and the descending section becomes slower and more significant with the increase in age.

The peak stress and peak strain of shotcrete at different curing temperatures are shown in Table 7. High temperature curing is not conducive to the development of peak stress of shotcrete, and the decrease at 40 is the most serious. It can be found that the peak strain of shotcrete does not change significantly at the age of 7 days and curing at 40 °C and 60 °C. At the age of 28 days, the peak strain of shotcrete cured at 40 °C increased by 8%, while that of shotcrete cured at 60 °C decreased by 16%. With the extension of age, the peak strain of shotcrete under high temperature curing is lower than that under standard curing.

#### 3.2.3. Effect of Basalt Fiber on Stress-Strain Curve of Shotcrete

The stress-strain curves of BFRS with different content cured at 60 °C are shown in Figure 6. It can be seen that an appropriate amount of BF could reduce the brittleness of shotcrete to a certain extent and make the stress-strain curve flatter and wider.

At the age of 7 days, the incorporation of moderate fiber can make the stress–strain curve of BFRS wider and flatter while the slope of the upward section becomes smaller. This is the reason why the peak value shifts to the right and decreases slightly. At the age of 28 days, the effect of fiber shows a similar phenomenon to day 7. After curing for 60 days, the peak point of the BFRS curve with the 0.3 and 0.4% content moves to the right, and the curve is smooth and full. However, the length of the BFRS curve with 0.1 and 0.2% content is obviously shorter, showing greater brittleness.

The peak stress and peak strain of shotcrete with fiber are shown in Table 8. It can be seen that the addition of fiber has little effect on the peak stress of shotcrete, and the peak stress of BFRS will decrease at later age. At the age of 60 days, when the content was 0.1%, 0.2% and 0.3%, 0.4%, the peak stress of BFRS decreased by 37.8%, 42.17% and 18.12%, 12.34%, respectively. Fiber can obviously increase the peak strain of shotcrete, and this phenomenon becomes more prominent along with the increase in fiber content and curing time.

At the age of 7 days, the peak strain of BFRS is higher than that of reference, and 0.3% attains the maximum of 91.6%. At the age of 28 days, the peak strain of the 0.1% and 0.2% increased by 26.2% and 42.8%, respectively. After being cured for 60 days, less BF will lead to a decrease in peak strain of BFRS, while the peak strain of the 0.3% and 0.4% increased by 68.8% and 58.3%, respectively. It can be seen that high temperature can significantly improve the peak strain of concrete. When damaged hardened paste loses its bearing capacity, fiber can partly bear capacity and restrict the deformation of surrounding rock [24,25], resulting in the enhancement of concrete toughness [26].

## 4. Constitutive Model

After comparative analysis, this paper selects the concrete compressive stress-strain constitutive relation proposed by Professor Guo as polynomial and rational, as shown in the following formula [27,28].
(2)$$\{\begin{array}{c} y=ax+\left(3-2a\right)x^{2}+\left(a-2\right)x^{3}\;(0\leq x\leq 1)\\ y=\frac{x}{b{\left(x-1\right)}^{2}+x}\;\left(x\geq 1\right) \end{array}$$
where *x* = *ε/ε*_p,t,_ *y* = *σ/σ*_p,t_, σ,ε  is stress and strain of concrete in a high geothermal environment, respectively. a, b are the parameters of the ascending section and descending section of the concrete stress-strain curve. Generally, the smaller the a value is, the higher the b value is, which indicates that that the curve is steeper and the area under the curve is smaller and smaller, indicating that the plastic deformation of concrete is small, the residual strength is low, the failure process is rapid, and the brittleness of concrete is larger [29,30].

According to Equation (1), the stress-strain curve of concrete is normalized [31,32], and the full stress-strain fitting curve of concrete and the corresponding parameter values are obtained.

### 4.1. Stress-Strain Curve Fitting of Shotcrete at Different Temperatures

The stress-strain fitting curve of shotcrete at different temperatures is shown in Figure 7 and the parameters in the ascending and descending section are presented in Table 9. It can be seen that the normalized concrete stress-strain curve has similar geometric characteristics to the measured curve. Meanwhile, it can be indicated that high temperature is beneficial to the increase in a value of shotcrete. This indicates that, before the concrete reaches the peak load, high temperature curing can increase the load of shotcrete and the stress-strain curve rises continuously. However, compared with the standard curing, the b value of shotcrete cured at a high temperature is higher, implying that partial damage in the interface area of concrete may be generated due to the excessive temperature. So, after reaching the maximum load, the crack resistance of concrete is insufficient and therefore concrete is destroyed rapidly. High temperature has little effect on the a value of shotcrete with 0.3% BF, while it contributes to the decrease in b value in the early stage. Compared with the specimen cured at standard conditions, the b value at day 7 at 40 °C and 60 °C decreases by 34.7% and 32% respectively. With the increase in age, the b value of shotcrete at high temperature increases.

### 4.2. Stress-Strain Curve Fitting of Shotcrete with Different Basalt Fiber Content

The stress-strain fitting curve of fiber shotcrete with different contents is shown in Figure 8, and the curve parameters are shown in Table 10. It can be seen that moderate fiber can increase the a value, by 17.4% at 7 days and 27.4% at 28 days, respectively. While fiber has a great influence on the b value of shotcrete at early age, especially at the age of 7 days, the b value of shotcrete with fiber is lower than that without BF, and when the addition of BF is 0.2%, the decrease in b value attains a maximum of 50.8%.

With the increase in age, the fiber has little effect on b value. This is the reason why the lower circumference area of BFRS in the curve becomes larger, indicating the energy dissipation effect of BF in shotcrete will improve the toughness and result in better plastic ability [33].

### 4.3. Establishment of Constitutive Equation

By fitting the ascending and descending sections of the stress-strain curve of shotcrete without fiber at 60 days, the relationship between the curing temperature of shotcrete and the parameters a and b of the ascending section and descending section can be obtained, as shown in Equation (3).
(3)$$a=0.05005T-0.23933\;\;R^{2}=0.989;b=0.00945T+1.61467\;R^{2}=0.957$$

In order to obtain the influence of basalt fiber content on the parameters a and b in ascending and descending sections, the influence coefficient λ of basalt fiber content is introduced into Equation (3), and Equation (4) is obtained. The influence coefficient λ of basalt fiber content is obtained from the parameters of the ascending section and descending section of BF-0.1, BF-0.2, BF-0.3, and BF-0.4 in Table 7.
(4)$$a=0.05005\lambda T-0.23933\lambda \;\;R^{2}=0.977;\;b=0.00945\lambda T+1.61467\lambda \;\;R^{2}=0.948$$

The stress-strain constitutive equation of BFRC in a high geothermal environment can be obtained by combining Equation (4) into (2), as shown in Equation (5).
(5)$$\{\begin{array}{c} y=\left(0.05005\lambda T-0.23933\lambda \right)x+[3-2\left(0.05005\lambda T-0.23933\lambda \right)]x^{2}+[\left(0.05005\lambda T-0.23933\lambda \right)-2]x^{3},0\leq \;x\leq 1\\ y=\frac{x}{\left(0.00945\lambda T+1.61467\lambda \right)\;{\left(x-1\right)}^{2}+x},\;\;x\geq 1 \end{array}$$
where *x* = *ε*/*ε_p_*_,*t*,_ *y* = *σ*/*σ_p_*_,*t,*_ and *ε* and *σ* are the stress and strain of basalt fiber shotcrete, respectively.

## 5. Conclusions

In this paper, the effects of high temperature and fiber content on the strength and uniaxial compression constitutive relation of basalt fiber reinforced shotcrete under high ground temperature are studied. The main conclusions are as follows:(1)High temperature curing can improve the early compressive strength of shotcrete, but it is not good for the development of its early flexural strength. After curing for 28 days, both the compressive strength and flexural strength decrease to varying degrees. The addition of fiber can reduce the strength deterioration of shotcrete caused by high temperature environment conservation, and the effect increases first and then decreases with the addition of fiber. The optimum basalt fiber content is 0.3%.(2)High temperature is attributed to the decrease in the peak stress and strain of shotcrete, and showing a greater brittleness, and the stress-strain curve becomes sharp and steep. Meanwhile, the fiber increases the peak strain of concrete, up to 91.6%, resulting in an increase in the plasticity and ductility.(3)According to the relationship between peak stress, peak strain, and axial compression, the stress-strain curve parameters a, b and temperature and basalt fiber content of concrete were analyzed, and the constitutive equation of shotcrete in a high geothermal environment is established.

## 6. Outlook

At present, the researchers of shotcrete in high geothermal environments are more focused on the material and structure level, and the hydration process and mechanism of shotcrete in high geothermal environments need to be further studied. In addition, it is necessary to carry out in-depth research on the damage mechanism and influence law of mechanical properties and durability of shotcrete under the actions of temperature, surrounding rock pressure, and erosion ion coupling, not only staying at the level of temperature influence, but also to provide support for tunnel durability design.

## Figures and Tables

**Figure 1 materials-14-07816-f001:**
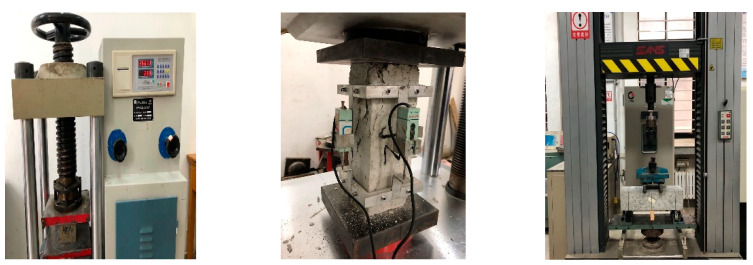
Mechanical property test.

**Figure 2 materials-14-07816-f002:**
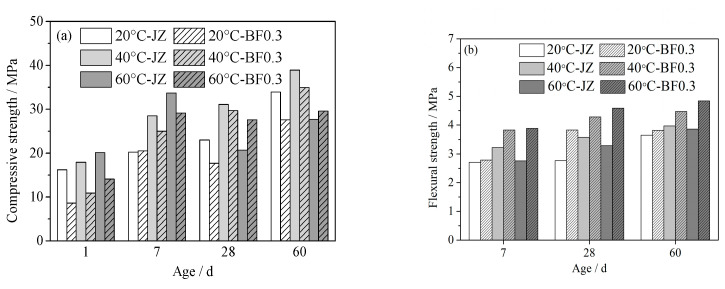
Effect of temperature on the strength of shotcrete: (**a**) compressive strength, (**b**) flexural strength.

**Figure 3 materials-14-07816-f003:**
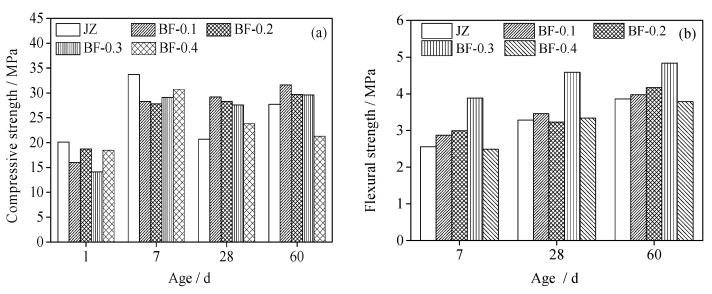
The effect of basalt fiber content on the strength of shotcrete at 60 °C: (**a**) compressive strength, (**b**) Flexural strength.

**Figure 4 materials-14-07816-f004:**
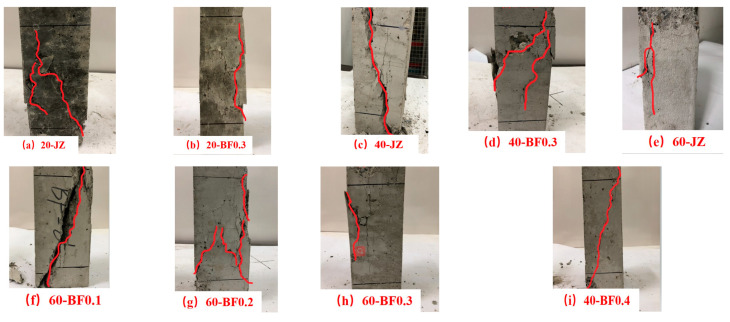
Effect of temperature and fiber content on failure pattern of axial compression of shotcrete.

**Figure 5 materials-14-07816-f005:**
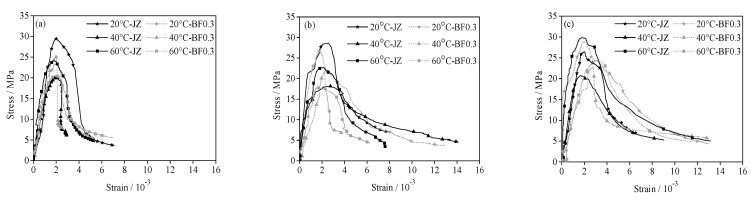
Effect of curing temperature on the stress-strain curve of shotcrete: (**a**) 7 d, (**b**) 28 d, (**c**) 60 d.

**Figure 6 materials-14-07816-f006:**
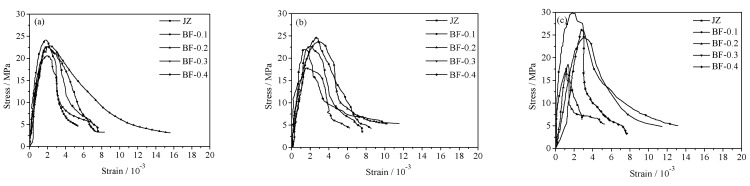
Effect of basalt fiber volume content on stress-strain curve of shotcrete at 60 °C. (**a**) 7 d, (**b**) 28 d, (**c**) 60 d.

**Figure 7 materials-14-07816-f007:**
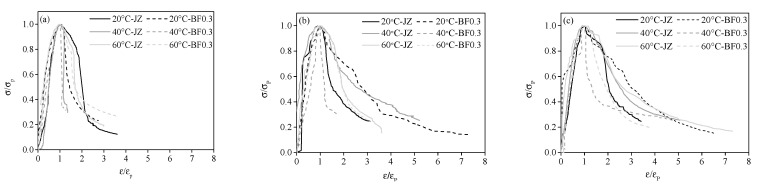
Stress-strain curve of shotcrete at different curing temperatures (**a**) 7 d, (**b**) 28 d, (**c**) 60 d.

**Figure 8 materials-14-07816-f008:**
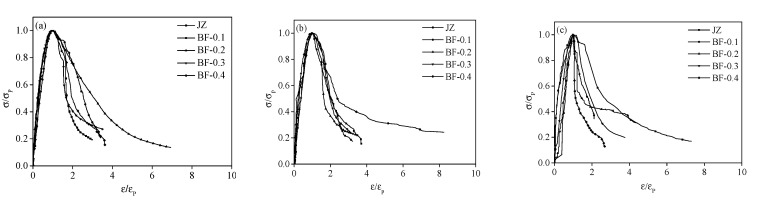
Stress-strain curve fitting of shotcrete at different basalt fiber content at 60 °C: (**a**) 7 d, (**b**) 28 d, (**c**) 60 d.

**Table 1 materials-14-07816-t001:** Chemical composition of OPC.

Chemical Composition (wt.%)	CaO	SiO_2_	Al_2_O_3_	Fe_2_O_3_	Na_2_O	K_2_O	MgO	SO_3_	TiO_2_	Loss on Ignition
Value	61.83	19.68	4.72	3.66	0.43	1.27	1.31	0.05	2.73	2.23

**Table 2 materials-14-07816-t002:** Physical properties of OPC.

Physical Properties	Initial Setting Time/Min	Final Setting Time/Min	Specific Surface (m^2^/kg)	Flexural Strength /MPa		Compressive Strength /MPa	
				3 d	28 d	3 d	28 d
Value	135	225	325	3.9	7.1	19.8	44.6

**Table 3 materials-14-07816-t003:** Physical properties of aggregate.

Physical Properties	Particle Size/mm	Bulk Density (g/cm^3^)	Apparent Density (g/cm^3^)	Clay Content/%	Crush Value/%	Water Absorption/%	Fineness Modulus
Stone	5–10	1.43	2.82	0.75	5	1.02	-
Sand	0–4.75	1.48	2.63	1.0	-	1.2	2.7

**Table 4 materials-14-07816-t004:** Chemical compositions of accelerator/%.

Chemical Composition	Al_2_O_3_	SO_4_^2−^	Na_2_O	Al_2_O_3_/SO_4_^2^^−^	pH (20 °C)
Value	11.7	25.6	-	0.46	3.7

**Table 5 materials-14-07816-t005:** Properties of basalt fiber.

Length /mm	Diameter /μm	Density (g/cm^3^)	Modulus of Elasticity /GPa	Elongation /%	Tensile Strength /MPa
6	15	2.56	40	3.1	2400

**Table 6 materials-14-07816-t006:** Mix proportion of reference concrete/(kg/m^3^).

Cement	Sand	Gravel	Fiber	Water	Accelerator	Superplasticizer
400	813	870	0	190	16	4
400	813	870	0.4	190	16	4
400	813	870	0.8	190	16	4
400	813	870	1.2	190	16	4
400	813	870	1.6	190	16	4

**Table 7 materials-14-07816-t007:** Peak stress and peak strain of common shotcrete under different curing temperatures.

Condition Symbols	*σ* _p_			*ε* _p_		
	7 d	28 d	60 d	7 d	28 d	60 d
20 °C-JZ	29.47	28.59	26.51	2.01	2.5	1.98
40 °C-JZ	20.12	18.16	20.70	1.98	2.70	1.65
60 °C-JZ	23.68	22.67	29.84	2.05	2.10	1.81

**Table 8 materials-14-07816-t008:** Peak stress and peak strain of shotcrete with different amounts of basalt fiber.

Condition Symbols	*σ* _p_			*ε* _p_		
	7 d	28 d	60 d	7 d	28 d	60 d
JZ	24.14	22.67	29.84	1.80	2.10	1.70
BF-0.1	22.65	24.62	18.55	2.01	2.65	1.35
BF-0.2	22.82	23.72	17.26	2.25	3.0	1.20
BF-0.3	20.58	17.81	24.43	3.45	1.55	3.04
BF-0.4	22.67	21.98	26.16	2.05	1.40	2.85

**Table 9 materials-14-07816-t009:** The parameters of the ascending and descending section of the full stress-strain curve of shotcrete at different curing temperatures.

Temperature/°C	Mixture	Ascending Section Parameter a			Descending Section Parameter b		
		7 d	28 d	60 d	7 d	28 d	60 d
20	JZ	0.821	2.090	0.772	0.288	2.268	1.551
	BF-0.3	2.173	1.580	2.723	3.459	2.604	2.113
40	JZ	0.954	2.953	1.742	5.754	2.643	2.498
	BF-0.3	0.961	1.757	0.923	2.260	3.545	3.714
60	JZ	2.238	2.103	2.774	3.132	2.844	1.929
	BF-0.3	1.872	1.306	1.275	2.350	2.148	2.383

**Table 10 materials-14-07816-t010:** The parameters of the ascending and descending sections of the full stress-strain curve of shotcrete at basalt fiber content.

Fiber Content/%	Ascending Section Parameter a			Descending Section Parameter b		
	7 d	28 d	60 d	7 d	28 d	60 d
JZ	2.238	2.103	2.174	3.132	2.844	1.929
BF-0.1	1.531	1.567	2.705	1.540	2.457	2.334
BF-0.2	2.628	2.666	0.227	2.206	3.257	2.512
BF-0.3	1.872	1.306	1.275	2.350	2.148	2.383
BF-0.4	2.237	2.139	1.077	2.644	1.95	3.178

## Data Availability

Authors have checked the data and comfirmed that no ethics issue exist in this paper.

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
