# Peer review of "Study on Mechanical Properties of Basalt Fiber Shotcrete in High Geothermal Environment"

_materials, 2021, doi:10.3390/ma14247816_

Round 1

Reviewer 1 Report

The present article investigates the effects of high temperature and fiber content on the strength and uniaxial compression of basalt fiber reinforced shotcrete.  The manuscript well organized and written. However, there are some points should be considered before publication as follow:

  • The title is not releveant to the content of the paperand it can be changed as the effect of basalt fiber on the mechanichal properties of shotcrete under high temperature.
  • The test conditions and setup of experiment needs to be explained clearly.
  • The effect of basalt fiber was not reported on the elastic region and it seems that the maximum strength in elastic region ( module of elasticity) does not change significantly. It is recomneded that the effect of reinforment should be highlighted in this region.  The crack growth is fast in brittle material and after reaching the maximum strength the material is considered as the failed sample. Therefore it is more interesting to show much cleary the effect of reinforment in the elastic 

After modification of the abovementioned comments the manuscript can be published.

Author Response

Please note our response is in blue in the revised manuscript.

Reviewer #1

1.The title is not releveant to the content of the paperand it can be changed as the effect of basalt fiber on the mechanichal properties of shotcrete under high temperature.

--- This has been revised in the manuscript.

2.The test conditions and setup of experiment needs to be explained clearly.

--- The detailed setups has been supplied into the manuscript.

3.The effect of basalt fiber was not reported on the elastic region and it seems that the maximum strength in elastic region ( module of elasticity) does not change significantly. It is recomneded that the effect of reinforment should be highlighted in this region. The crack growth is fast in brittle material and after reaching the maximum strength the material is considered as the failed sample. Therefore it is more interesting to show much cleary the effect of reinforment in the elastic.

--- This interpretation into the manuscript by referring to some literature.

Reviewer 2 Report

This article in three different temperature (20/40/60) and 55% relative humidity conditions, the simulation environment of basalt fiber reinforced shotcrete strength and the influence of the stress-strain curve, and set up variables for sprayed concrete constitutive model of fiber content and temperature. With the increase of temperature, the mechanical properties of BFRS improved in the early stage, but decreased in the later stage. The addition of basalt greatly improved the mechanical properties of shotcrete. The article lists a lot of tables and pictures and is very organized and the logic of the article is strong, and the content before and after corresponds to each other.

However, there are still some issues to be addressed. A minor revision is suggested before its acceptance.

  1. The abstract part lacks key data and does not summarize the whole paper well.
  2. Many of the abbreviations in the article are not explained.
  3. The relevant background and references of basalt fibers should be appropriately added.
  4. The conclusion part only summarizes the experimental content and cannot be called the conclusion.
  5. The future application of this work should be prospected.
  6. The quality of the references is low and some of them are too old. Some other bio-derived fiber reinforced composites should be carefully read and mentioned: https://doi.org/10.1016/j.jobab.2021.11.002; https://doi.org/10.1016/j.jobab.2020.03.003; https://doi.org/10.1016/j.jobab.2021.04.004
  7. There are some spelling and grammar problems in the manuscript. English editing service should be performed before submission.

Author Response

Please note our response is in green in the revised manuscript.

Reviewer #2

1.The abstract part lacks key data and does not summarize the whole paper well.

--- This has been revised in the manuscript.

2.Many of the abbreviations in the article are not explained.

--- This has been rechecked and revised in the manuscript.

3.The relevant background and references of basalt fibers should be appropriately added.

--- In the introduction, authors intend to make a statement on the research status of the application of different fiber at high geothermal environment. However, there is little literature on the application of basalt fiber at the specific condition. So, the relevant background and references of basalt fiber is insufficient and this is also the focus of this paper.

4.The conclusion part only summarizes the experimental content and cannot be called the conclusion.

--- This has been revised in the manuscript.

5.The future application of this work should be prospected.

--- This has been supplied in the manuscript.

6.The quality of the references is low and some of them are too old. Some other bio-derived fiber reinforced composites should be carefully read and mentioned: https://doi.org/10.1016/j.jobab.2021.11.002; https://doi.org/10.1016/j.jobab.2020.03.003; https://doi.org/10.1016/j.jobab.2021.04.004

--- It has been quoted in the manuscript.

7.There are some spelling and grammar problems in the manuscript. English editing service should be performed before submission.

--- The language has been polished.

Reviewer 3 Report

The research article "Stress-strain behavior of basalt fiber reinforced shotcrete in the high geothermal environment" is well written.

1. There is a further need to highlight the importance of this study in the introduction section.

2. Table 1: OPC is presented two times, what is the difference between them.

3. Table 4, there are no specific designations for different mixes.

4. Figure 3. It's better to add labels for each figure.

5. Please explain failure modes in more detail.

6. Results need some more discussion and comparison with the existing studies.

Author Response

Please note our response is in yellow in the revised manuscript.

Reviewer #3

  1. There is a further need to highlight the importance of this study in the introduction section.

--- This has been revised in the manuscript.

  1. Table 1: OPC is presented two times, what is the difference between them.

--- No difference exists between them and all properties are specific to the same OPC. The Table 1 has been separated into three tables to let the content better understood.

  1. Table 4, there are no specific designations for different mixes.

--- This has been revised in the manuscript.

  1. Figure 3. It's better to add labels for each figure.

--- This has been revised in the manuscript.

  1. Please explain failure modes in more detail.

--- A detailed explanation has been supplied in the manuscript.

  1. Results need some more discussion and comparison with the existing studies.

--- This has been supplied in the manuscript.

This manuscript is a resubmission of an earlier submission. The following is a list of the peer review reports and author responses from that submission.

Round 1

Reviewer 1 Report

The submitted paper is well written and well organized, however, it has no contribution with the Polymeric materials and it is out of the scope of this journal. Hence I have to reject this manuscript for publication.

Reviewer 2 Report

Comments

This article in three different temperature (20/40/60) and 55% relative humidity conditions, the simulation environment of basalt fiber reinforced shotcrete strength and the influence of the stress-strain curve, and set up variables for sprayed concrete constitutive model of fiber content and temperature. With the increase of temperature, the mechanical properties of BFRS improved in the early stage, but decreased in the later stage. The addition of basalt greatly improved the mechanical properties of shotcrete. The article lists a lot of tables and pictures and is very organized and the logic of the article is strong, and the content before and after corresponds to each other.

However, there are still some issues to be addressed. A minor revision is suggested before its acceptance.

  1. The abstract part lacks key data and does not summarize the whole paper well.
  2. Many of the abbreviations in the article are not explained.
  3. The relevant background and references of basalt fibers should be appropriately added.
  4. The conclusion part only summarizes the experimental content and cannot be called the conclusion.
  5. The future application of this work should be prospected.
  6. The quality of the references is low and some of them are too old. Some other bio-derived fiber reinforced composites should be carefully read and mentioned: https://doi.org/10.1016/j.jobab.2021.11.002; https://doi.org/10.1016/j.jobab.2020.03.003; https://doi.org/10.1016/j.jobab.2021.04.004
  7. There are some spelling and grammar problems in the manuscript. English editing service should be performed before submission.

Reviewer 3 Report

The research article "Stress-strain behavior of basalt fiber reinforced shotcrete in high geothermal environment" is well written.

There is further neeed to highlight the importance of this study in the introduction section.

Table 1: OPC is presented two times, what is difference between them.

Table 4, there are no specific designations for different mixes.

Figure 3. Its better to add labes for each figure.

Results need some more discussion and comparison with the existing studies.